# NS1-mediated DNMT1 degradation regulates human bocavirus 1 replication and RNA processing

Shuangkang Qin[1,2☯], Honghe Chen[1☯¤], Chuchu Tian[1], Zhen Chen[1], Li Zuo[1,2], Xueyan Zhang[1], Haojie Hao[1,3], Fang Huang[3], Haibin Liu[1,3]*, Xiulian Sun[1]*, Wuxiang Guan [1,3]*

1 Center for Emerging Infectious Diseases, Wuhan Institute of Virology, Center for Biosafety Mega-Science, Chinese Academy of Sciences, Wuhan, Hubei, China, 2 University of Chinese Academy of Sciences, Beijing, China, 3 Hubei JiangXia Laboratory, Wuhan, Hubei, China

☯ These authors contributed equally to this work.
¤ Current address: National Research Institute of Food & Fermentation Industries Corporation Limited, Beijing, China
* hbliu@wh.iov.cn (HL); sunxl@wh.iov.cn (XS); guanwx@wh.iov.cn (WG)

**Data Availability Statement:** All data are in the manuscript and/or supporting information files.

**Funding:** This study was supported by the Strategic Priority Research Program of the Chinese

## Abstract

Methylation of the DNA genome plays an important role in viral gene inactivation. However, the role of DNA methylation in human bocavirus (HBoV) remains unclear. In this study, the HBoV1 genomic DNA was found extensively methylated at the CHG and CHH sites. Inhibiting DNA methylation with 5-aza-2′-deoxycytidine (DAC) altered the methylation status and reduced viral DNA production, while enhanced the RNA splicing at D1 and D3 sites and the polyadenylation at the proximal polyadenylation site, (pA)p. Knockdown of DNA methyltransferase 1 (DNMT1) had the same effect on viral DNA synthesis and RNA processing as the DAC treatment, indicating that DNMT1 is the major host methyltransferase involved in viral DNA methylation. In addition, the nonstructural protein NS1 promoted DNMT1 degradation through the ubiquitin-proteasome pathway to regulate viral replication and RNA processing. Collectively, the results suggest that DNA methylation and DNMT1 facilitate HBoV replication and are essential for appropriate NS1 localization in the nucleus. DNMT1 degradation through NS1 promotes the virus RNA processing, leading to viral protein expression.

## Author summary

DNA methylation is an important epigenetic mark involved in both host biological processes and virus replication cycle. The mechanism underlying the precisely organized viral DNA replication and RNA processing in parvovirus infection is still largely unknown. This study illustrates that these processes are regulated by DNMT1 mediated epigenetic mechanisms. DNA methylation and DNMT1 facilitate HBoV DNA replication but repress its RNA processing. NS1, besides the initiation of DNA replication, promotes the viral RNA processing through DNMT1 degradation by the proteasome pathway. Our

Academy of Sciences [XDB0490000 to WG],
National Natural Science Foundation of China
[31970168 to WG] and [31400780 to HC], Key
R&D Program of Hubei Province [2021BCD004 to
WG], Hubei Science and Technology Major Project
[2021ACB004 to WG], and Emergency Key Project
of Guangzhou Laboratory [EKPG21-30-2 to WG].
The funders had no role in study design, data
collection and analysis, decision to publish, or
preparation of the manuscript.

**Competing interests:** The authors have declared
that no competing interests exist.

finding also indicates that the DNMT1 mediated DNA methylation system is a promising
target for preventing HBoV infection.

## Introduction

Human bocavirus (HBoV) was first isolated from children's nasopharyngeal aspirates in 2005
[1]. It is associated with upper or lower respiratory tract diseases in young children and is
often associated with other respiratory viruses [2–4]. HBoV belongs to the genus *Bocaparvo-
virus* within the family *Parvoviridae* [1,5]. It is a small, non-enveloped, single-stranded DNA
virus containing a 5543 nt linear genome flanked by terminal palindromes. HBoV RNA is
transcribed from the P5 promoter, processed into different isoforms by alternative splicing,
and terminated by two alternative polyadenylation cleavage sites. These mRNAs are translated
into at least six non-structural proteins, NS1, NS2, NS3, NS4, NS1-70, and NP1 [6–8] and
three structural proteins, VP1, VP2, and VP3 [6,7,9]. Among them, NS1 binds to the viral rep-
lication origin to initiate viral DNA replication and also interacts with various cellular proteins
to regulate this process [10–15]. NS1 contains three functional domains, the DNA-binding
domain at N terminus, the helicase domain in the middle and the transactivation domain at C
terminus. NP1 is exclusively encoded by bocaviruses and is involved in viral RNA processing.
It promotes the use of the D3 donor site and the distal polyadenylation site, (pA)d, which is
required for capsid protein production [6,16–21].

5-Methylcytosine (5mC) is the major DNA methylation molecule in mammals, which is
generated by the transfer of a methyl group to the C5 position of cytosine and catalyzed by
DNA methyltransferases (DNMTs) such as DNMT1, DNMT3a, and DNMT3b [22]. It plays
an important role in various cellular and developmental processes [23,24] as methylated DNA
alters DNA conformation and transcription factor binding to regulate gene expression [25].
DNA methylation has been identified in various DNA viruses, including adenovirus, human
papillomavirus (HPV), Epstein-Barr virus (EBV), hepatitis B virus (HBV), and retroviruses
[26–30]. Promoter methylation results in viral gene silencing in adenoviruses, EBV, and HBV
[28,31–33], and methylation of CpG islands flanking the HIV-1 transcription start site is
required for HIV-1 latency [34]. CpG methylation in parvovirus B19 is found in clinical sam-
ples and infected cell cultures, and is correlated with low viral gene expression [35,36]. How-
ever, the role of DNA methylation in parvoviruses is still poorly understood.

DNA replication and RNA processing in parvoviruses are delicately orchestrated and regu-
lated after the uncoated ssDNA genome enters the host nucleus [37]. ssDNA needs to be con-
verted to dsDNA intermediates using host cellular machinery and then transcribed into
mRNA of viral regulatory proteins NS1 and NP1 prior to viral DNA replication, as NS1 is
essential for the recognition of the replication origin and initiation of DNA replication [37,38].
Once viral DNA replicates to an appropriate level, it is packaged into capsid proteins for viral
assembly. In many other DNA viruses, capsid (structural) proteins are encoded by a class of
late genes that are separately transcribed from distinct promoters after viral DNA synthesis
[38,39]. However, parvovirus has only one promoter and transcribes a single viral RNA pre-
cursor to generate multiple mRNAs for all viral proteins [37]. Although NP1 has been charac-
terized to regulate viral RNA processing for capsid protein production [17,19,20], whether
DNA methylation are involved in parvoviruses DNA replication and RNA processing is still
largely unknown.

The aim of this study was to characterize the roles of DNA methylation and DNMT1 in
HBoV infection. We demonstrated that DNMT1 mediated DNA methylation regulated both

virus DNA replication and RNA processing. In particular, DNA methylation suppressed the viral RNA splicing at D3 site and the polyadenylation at (pA)p, which was important for the viral protein expression. Furthermore, a new function of NS1 in the regulation of viral RNA splicing and polyadenylation by promoting DNMT1 degradation was discovered. The results suggest that HBoV hijacks the host methyltransferase DNMT1 to complete its replication cycles.

## Results

### HBoV genomic DNAs are methylated at CHG and CHH sites

To investigate DNA methylation in the HBoV genome, whole-genome bisulfite sequencing was performed on HEK293T cells transfected with the HBoV infectious clone and 37 5mCs were identified in the HBoV genome (Table 1). Among them, two 5mCs were in 3' UTR, 13 in the NS1coding region, 5 in the NP1 coding region, and 17 in the VP coding region (Table 1 and Fig 1A). Two methylation patterns, CHG and CHH (where H stands for A, T, or C), instead of cytosine guanine dinucleotide (CpG), were discovered at methylation sites. To further confirm the methylation sites were not from bacterial cultured plasmid, the plasmid DNA was removed by DpnI digestion before whole genome bisulfite sequencing. The most methylation sites were consistent with the identified sites without DpnI digestion (Fig 1A and S1 Table). Transfer of the methyl group to 5mCs requires host DNMTs. 5-aza-2'-deoxycytidine (DAC), which is a cytosine nucleoside analog trapping DNMTs, was used to inhibit HBoV genome methylation. DAC treatment induced a general decrease in DNMTs in HBoV DNA-transfected cells, suggesting efficient inhibition of these enzymes (Fig 1B). We observed that a few 5mCs were altered in DAC treated samples (Fig 1A, bottom). To investigate whether DNMTs were dysregulated, the expression levels were evaluated in HBoV-transfected cells. The abundance of DNMT1 remained unchanged up to 24 hours post-transfection but decreased dramatically by 48 hours (Fig 1C). In contrast, the expression levels of DNMT2, DNMT3A, DNMT3B, and DNMT3L remained unaffected (Fig 1C). To further confirm the altered DNMT1 expression by HBoV infection, the differentiated human bronchial epithelial Calu-3 cells were prepared for virus infection at a multiplicity of infection (MOI) of 100 genome copy numbers/cell as previously reported [40]. The viral infection was associated with low DNMT1 expression at 72 h and 120 h post-infection (Fig 1D). These results indicate that HBoV genomic DNA is methylated and that DNMT1 is downregulated by either HBoV infectious clone transfection or viral infection.

### DNA methylation is linked to viral replication and NS1 localization

As DNA viral genome methylation is an important player in viral infections [28,31–33], Southern blotting was performed to explore the effect of methylation inhibition on HBoV DNA production (Fig 2A). The signals of the DpnI-resistant bands, including the replicative form of HBoV DNA (RF), dimer of RF (dRF), and single-stranded DNA genome (ss), represented actively replicating DNA [41,42]. DAC treatment caused a four-fold decrease in DpnI-resistant bands (Fig 2A), indicating that HBoV methylation inhibition resulted in reduced viral DNA synthesis, which was further supported by more than two-fold decreases in Hirt DNA quantified as previously described [21] (Figs 2B and S1A). In contrast, the expression of mRNAs encoding NS1 and NP1 increased in the DAC-treated samples (Figs 2C and S1B). The changes in viral DNA replication and gene expression were not due to effects on the cell cycle caused by DAC treatment (S1C Fig).

To investigate why the enhanced NS1 and NP1 expressions were associated with reduced viral DNA synthesis after DAC treatment, the nuclear localizations of NS1 and NP1 were

**Table 1. HBoV1 methylation sites.**

| position | +/- chain | Cytosine pattern | Cytosine sequence context |
|---|---|---|---|
| 207 | + | CHG | CCCAG |
| 416 | + | CHH | TCCAT |
| 1157 | + | CHG | TCCAG |
| 1382 | + | CHG | ACCAG |
| 1637 | + | CHH | AACAA |
| 1765 | + | CHG | ACCAG |
| 2071 | + | CHH | AACAA |
| 2819 | + | CHG | TCCAG |
| 2983 | + | CHH | ATCAA |
| 2991 | + | CHG | TACAG |
| 3192 | + | CHG | GCCTG |
| 3525 | + | CHG | ACCAG |
| 4090 | + | CHG | ACCTG |
| 4314 | + | CHG | TCCTG |
| 4440 | + | CHG | ACCTG |
| 4872 | + | CHG | TCCAG |
| 5126 | + | CHG | ATCAG |
| 209 | - | CHG | CAGGA |
| 559 | - | CHH | AAGAT |
| 1159 | - | CHG | CAGGG |
| 1384 | - | CHG | CAGGA |
| 1682 | - | CHH | GTGTG |
| 1767 | - | CHG | CAGGA |
| 2078 | - | CHH | ATGGA |
| 2221 | - | CHG | CTGAT |
| 2821 | - | CHG | CAGGG |
| 2993 | - | CHG | CAGAA |
| 3194 | - | CHG | CTGGA |
| 3254 | - | CHG | CTGAT |
| 3527 | - | CHG | CAGGA |
| 4092 | - | CHG | CTGGA |
| 4316 | - | CHG | CTGGA |
| 4324 | - | CHH | ATGTT |
| 4442 | - | CHG | CTGGC |
| 4874 | - | CHG | CAGGC |
| 4894 | - | CHH | ATGGC |

examined by immunofluorescence staining. NP1 exclusively localized in the nucleus in the presence or absence of DAC treatment (Fig 2D). NS1 was localized in the nucleus in the absence DAC treatment (Fig 2E, top). However, the majority of NS1 was expressed and localized in the cytoplasm of DAC-treated cells in the presence of HBoV transfection (Fig 2E, bottom). Further staining of the NS1 localization at different time points indicated the translocation occurred after 24h post transfection (S1D Fig). The cytoplasmic localization of NS1, but not NP1, upon DAC treatment in HBoV transfection were further confirmed by subcellular fractionation experiments (Fig 2F and 2G). Collectively, HBoV methylation facilitates viral replication, altered viral gene expression and NS1 nuclear localization.

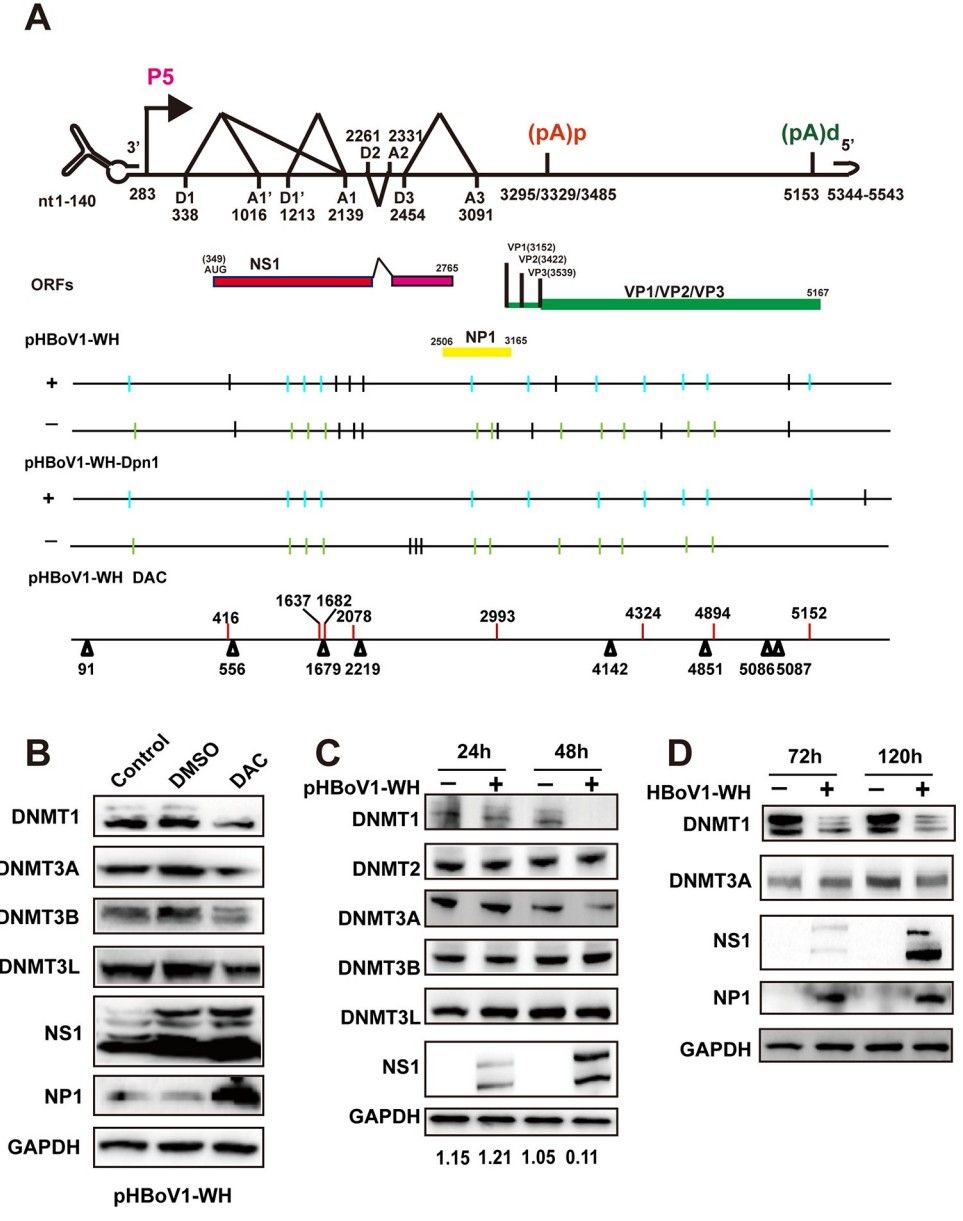

**Fig 1. HBoV genomic DNA is methylated by DNMTs.** (A) The HBoV genomic structure is diagramed with a P5 promoter, splicing donor sites (D1, D1', D2, and D3), splicing acceptor sites (A1, A1', A2, and A3), proximal polyadenylation site [(pA)p], and distal polyadenylation site [(pA)d]. The numbers represent the nucleotide positions. ORFs are diagramed under the genomic structure. The marked 5mCs were identified through bisulfite sequencing 48 h post-transfection. The methylation sites removed by the DAC treatment are labeled in red and new methylation sites are labeled in black. (B) HEK293T cells were transfected with the HBoV infectious clone (pHBoV1-WH) and treated with DMSO or 20uM DAC. The expressions of several DNA methyltransferases (DNMTs), NS1, and NP1 were analyzed through western blotting 24 h post-transfection. (C) The expression of DNMTs was analyzed 24 or 48 h post-transfection of pHBoV1-WH. (D) Calu-3 cells were infected with HBoV, and the expressions of DNMT1 and DNMT3A were analyzed 3d or 5d post-infection.

## DNA methylation suppresses HBoV RNA splicing and polyadenylation

DNA methylation is not only a transcriptional inactivator in host and viral transcription [28,31–33,43], but also regulates alternative RNA splicing [44] and polyadenylation [45]. We next investigated the effect of DAC treatment on these processes in HBoV transfection by

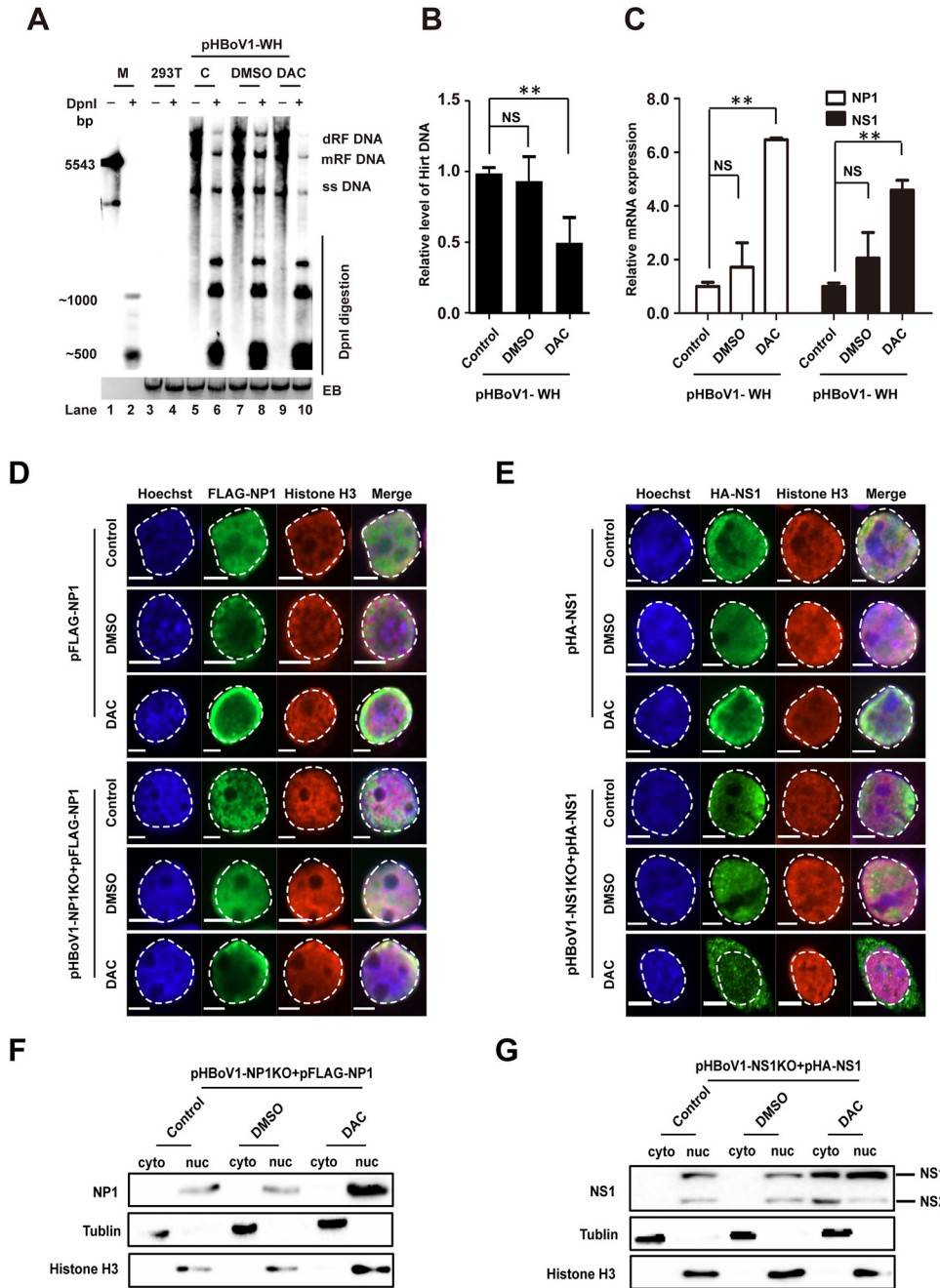

**Fig 2. DAC treatment represses viral replication associated with cytoplasmic NS1 translocation.** (A) Southern blotting. HEK293T cells were transfected with pHBoV1-WH and treated with DMSO or DAC. Hirt DNAs were extracted and digested with DpnI. The DNA was separated in 1% agarose gels and transferred to a Hybond-N$^+$ membrane, followed by hybridization with HBoV full-length probe. dRF, double replication form; mRF, monomer replication form. Un-transfected cells were used as control. (B) Dpn I digested Hirt DNAs in (A) were quantified and analyzed through qRT-PCR. Specific primers spanning DpnI digestion site were used in qRT-PCR, and GAPDH was used as a control **, p < 0.01. (C) The mRNAs expressing NS1 or NP1 in DAC-treated cells were quantified through qRT- PCR, and GAPDH was used as internal control. *, p < 0.05, **, p < 0.01. (D–E) The subcellular location of NS1 and NP1 in DAC-treated cells in immunofluorescence experiments. HEK293T cells were transfected with the indicated plasmids and treated with DAC or DMSO. Transfections of NS1 or NP1 mutated HBoV infectious clones (pHBoV-NP1KO or pHBoV-NS1KO) with FLAG tagged NP1 (FLAG-NP1) (D) or HA tagged NS1 (HA-NS1) (E) in the lower panel were performed, respectively, to visualize the NS1 or NP1 in the presence HBoV transfection. The localization of FLAG-NP1 (D) or HA-NS1 (E) was detected using specific antibodies against FLAG or HA (green) at 48h post transfection, respectively. The nuclei were stained with Hoechst 33258 (blue) or histone H3 specific antibody

(red). Scale bars, 5 μm. (F–G) Subcellular fractionation of HEK293T cells transfected with the indicated plasmids and treated with DAC or DMSO. FLAG-NP1 (F) or HA-NS1 (G) proteins in subcellular fractions were detected through western blot. The subcellular fractions of β-tublin and histone H3 indicated the efficient subcellular fractionation.

using northern blotting and an RNase protection assay (RPA) as described previously [41] (Fig 3A). Inhibition of HBoV methylation by DAC led to an approximately two-fold increase in HBoV RNA expression (Fig 3B), which was consistent with increased NP1 and NS1 mRNA levels (Fig 2C). Three probes targeting the donor sites D1 and D3 and the (pA)p cleavage site were used in RPA (Fig 3A). As shown in Fig 3C and 3D, more than three-fold viral RNAs were spliced at the D1 or D3 donor sites in DAC-treated samples compared to that in untreated samples. The polyadenylation at (pA)p increased more than two-fold (Fig 3E). The increases of viral RNA splicing and polyadenylation by DAC treatment imply that DNA methylation represses HBoV RNA processing.

## DNMT1 is the major methyltransferase in HBoV genomic DNA methylation

As DNMT1 is the major maintenance DNA methyltransferase in host cells [46] and is downregulated by HBoV infection (Fig 1C and 1D), we hypothesized that it is involved in DNA methylation-regulated viral replication and RNA processing. To this end, DNMT1 was efficiently knocked down by two shRNAs and the RNA processing was checked. Consistent with the DAC treatment, NS1 and NP1 expression was increased more than two-fold (Fig 4A) associated with cytoplasmic retention of NS1, but not NP1 (Fig 4D and 4E), which was not a result of an altered cell cycle (S1E Fig). The NS1 localization was further validated by subcellular fractionation experiments (S1F Fig). Notably, the viral DNA replication (Fig 4B) and the amount of Hirt DNA (Fig 4C) decreased at least a two-fold. DNMT1 knockdown also resulted in a two-fold increase in viral RNA expression (Fig 4F), a ten-fold increase in RNA splicing at the D1 (Fig 4G) or D3 donor sites (Fig 4H), and a three-fold increase of polyadenylation at (pA)p (Fig 4I). Collectively, these results showed that DNMT1 depletion strongly resembles the effect of the DAC treatment on HBoV transfection, indicating that DNMT1 is the major player involved in HBoV methylation.

## NS1 interacts with DNMT1 and promotes its degradation

DNMT1 is associated with the replication machinery and accumulates at host replication sites [47]. The interaction between DNMT1 and viral replication-related proteins NP1 and NS1 was investigated. DNMT1 did not co-immunoprecipitate with NP1 in the presence or absence of HBoV transfection (Fig 5A). In contrast, NS1 co-immunoprecipitated with the DNMT1 antibody in the presence or absence of HBoV transfection (Fig 5B), indicating that the interaction between NS1 and DNMT1 was independent of viral replication. Additionally, as the NS1 start codon-mutated virus, which did not express NS1 for viral DNA replication, failed to downregulate DNMT1, the downregulation of DNMT1 by HBoV transfection was dependent on NS1 (Fig 5C). Restoration of NS1 expression in the mutant virus or expression of NS1 alone had the similar negative effect on DNMT1 expression as the wild-type virus (Fig 5C). Furthermore, the proteasome inhibitor MG132 inhibited NS1 dependent downregulation of DNMT1, implying NS1 might promote DNMT1 degradation through the proteasome pathway (Fig 5D). Since K48-linked and K63-linked ubiquitination are the two most abundant ubiquitin types in mammalian cells, we checked whether NS1 regulated these ubiquitin-linked chains of DNMT1. DNMT1 was immunoprecipitated from ubiquitin, K48-ubiquitin, or K63-ubiquitin overexpressed cell lysates in the presence or absence of NS1, followed by

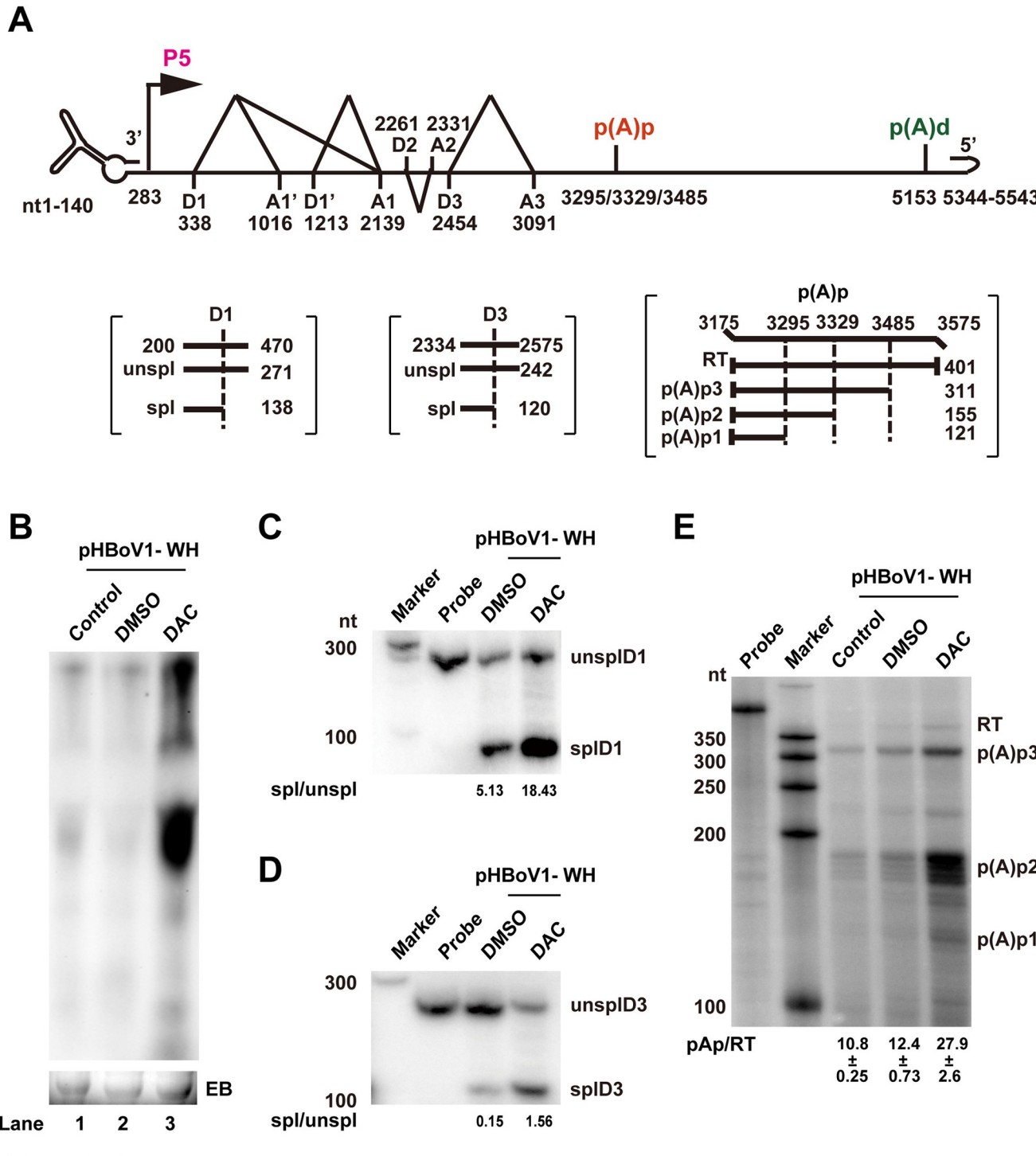

**Fig 3. DAC treatment regulates HBoV RNA processing.** (A) The top panel diagrams the HBoV genomic structure. The bottom panel shows the positions and sizes of the probes used in RPA. The size of the expected RPA products from unspliced RNA (unspl), spliced RNA (spl), and RNA using the (pA)p site are below each probe. RT, read through RNAs. (B) HBoV RNA expressions in DAC-treated cells were evaluated through northern blotting. HEK293T cells were transfected with pHBoV1-WH and treated with DAC for 48 h. Total RNAs (10 μg) prepared from transfected cells were resolved on 1.5% agarose gels, transferred to Hybond-N+-membranes and hybridized with probes spanning nt 349 to 5167. Ethidium bromide-stained 18S RNA bands were used as loading control. (C–E) RPA analysis of HBoV RNAs in DAC-treated cells. Total RNAs (10 μg) prepared from transfected HEK293T cells with DAC or DMSO treatment were protected with the indicated probes. HBoV RNAs spliced at D1 (C) or D3 (D) sites, or polyadenylated at (pA)p (E) were analyzed through RPA assays. The ratios of spliced to unspliced RNA, or polyadenylated RNA at (pA)p to read-through RNA (RT) were quantified and shown at the bottom.

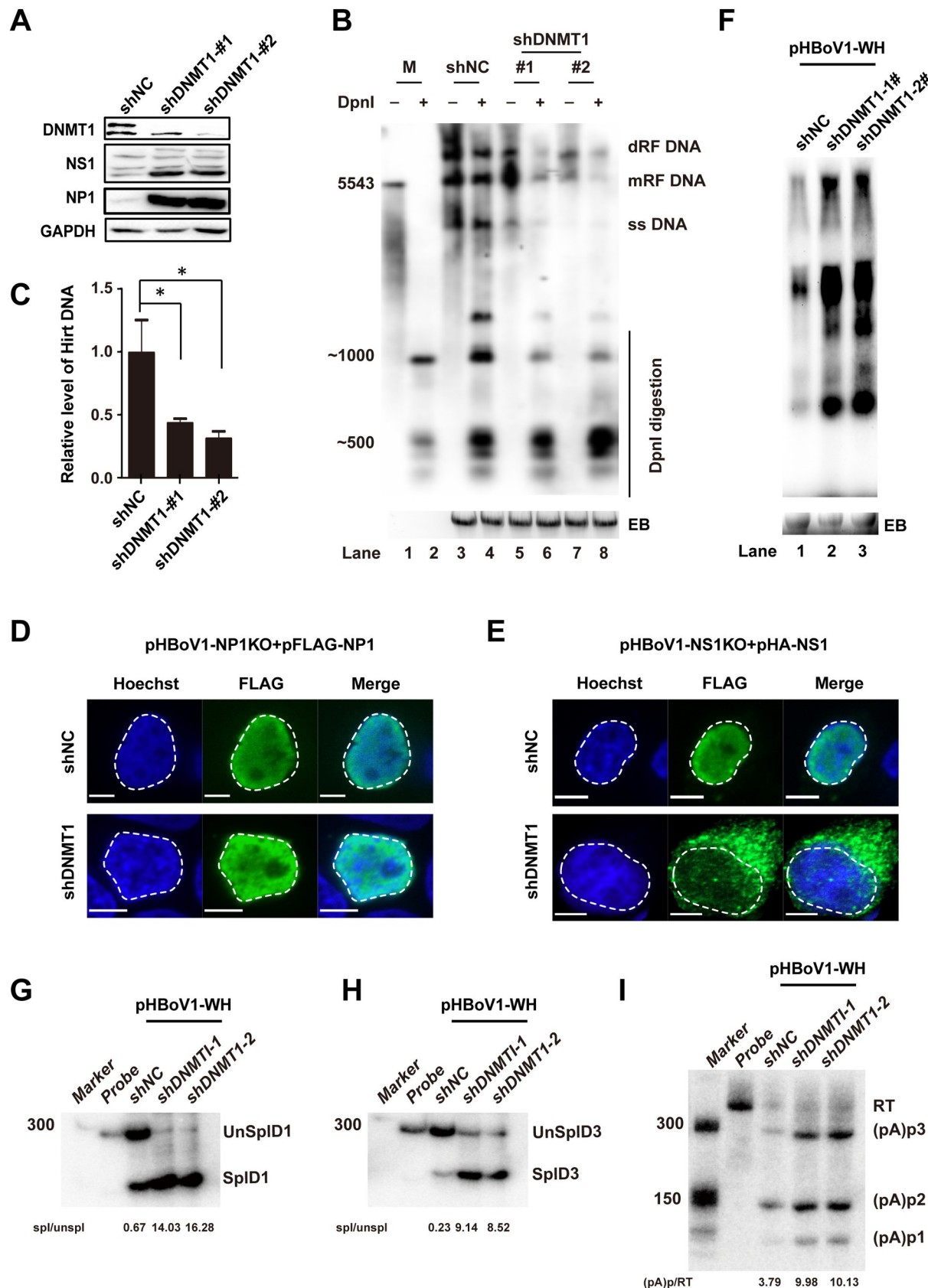

**Fig 4. Knockdown of DNMT1 strongly resembles the effects of DAC treatment on HBoV replication, NS1 localization, and RNA processing.** DNMT1 was knocked down by two shRNAs in HEK293T cells. Stably knockdown cells obtained using puromycin selection were transfected with pHBoV1-WH. (A) The NS1 or NP1 expression was quantified through western blotting, and GAPDH was used as loading control. Knockdown efficiencies were verified through western blotting. (B) HBoV replications were measured by Southern blotting as described above. (C) The reduced viral replication was further verified in DNMT1 knockdown cells through qRT-PCR quantification of Hirt DNA. *, p < 0.05. (D–E) The subcellular location of NS1 (D) and NP1 (E) in DNMT1-knockdown cells were detected by immunofluorescence as described in Fig 2D and 2E. Scale bars, 5 μm. (F–I) DNMT1 knockdown enhanced HBoV RNA processing. HBoV transcription in DNMT1-knockdown cells was evaluated through northern blotting (F). RPA analysis of HBoV RNAs spliced at D1 (G) or D3 (H) sites or polyadenylated at (pA)p (I) in DNMT1-kncokdown samples.

immunoblotting. Indeed, NS1 promoted K48-linked and K63-linked ubiquitination of DNMT1 (Fig 5E). These results demonstrate that NS1 promotes the degradation of DNMT1 through the ubiquitin-proteasome pathway.

HBoV produces several NS isoforms by alternative splicing [37]. To identify which specific isoforms interact with DNMT1 and mediate its degradation, cDNAs for four NS isoforms were cloned and expressed in HEK293T cells. NS1 is translated from D2-A2 spliced RNA, NS2 from D1'-A1 spliced RNA (lacking the helicase domain), NS3 from D1-A1' spliced RNA (lacking the DNA binding domain), and NS1-70 from D2-A2 retained RNA (which terminates at a stop codon in the intron region, resulting in a deletion of the transactivation domain) (Fig 6A). Since the NS1 cDNA included the intact D1'-A1 intron, it was spliced to produce the NS2 protein (Fig 6B). Despite similar levels of NS3 RNA compared to other isoforms, NS3 protein expression was very low (Fig 6B). NS1, NS2, and NS1-70 all efficiently mediated DNMT1 degradation (Fig 6C), and NS1 and NS2 directly interacted with DNMT1 (Fig 6D). Thus, while the transactivation domain is required for interaction with DNMT1, it is not necessary for NS1-mediated DNMT1 degradation.

Additionally, the recruitment of NS1 to the HBoV genome was reduced under DAC treatment, whereas the recruitment of Pol II was significantly enhanced (Fig 6E). These changes were consistent with the observed effects on viral DNA replication and RNA expression following DAC treatment (Fig 2A–2C). Taken together, DNMT1-mediated DNA methylation is crucial for viral replication, and NS1 interacts with this system to facilitate the viral life cycle.

## Discussion

Although DNA methylation in parvoviruses has been shown to correlate with low viral gene expression [35,36], its role in the HBoV life cycle remains unclear. In the present study, we found that 37 cytosines were methylated in two patterns, CHG and CHH, in the HBoV1 genome. HBoV 5mCs facilitate viral replication but repress viral RNA splicing, and polyadenylation at (pA)p. NS1 interacts with DNMT1, the major methyltransferase involved in HBoV genome methylation, and stimulates its degradation via the ubiquitin-proteasome pathway.

DNA methylation occurs predominantly at CpG sites in the promoter region [48]. However, the HBoV genome is methylated only at the CHG and CHH sites, which usually occur in plants and plant viruses and is catalyzed by specific chromomethylases [49,50]. Interestingly, extensive analysis of parvoviral genome methylation has shown high methylation capabilities in vertebrates, causing the depletion of CpG dinucleotides in the parvoviral genome [51]. Thus, we speculate that methylation occurs only at the CHG and CHH sites in the HBoV genome because of CpG depletion.

DNMT1, a component of the DNA replication complex, is required for host DNA replication and DNA methylation maintenance [52,53]. It binds to the nascent strand of DNA to copy the methylation pattern of the parental DNA [25]. Our results showed that inhibition (DAC treatment) or knockdown of DNMT1 resulted in decreased viral replication and cytoplasmic localization of the NS1 protein, although whether DAC treatment on the DNMT1

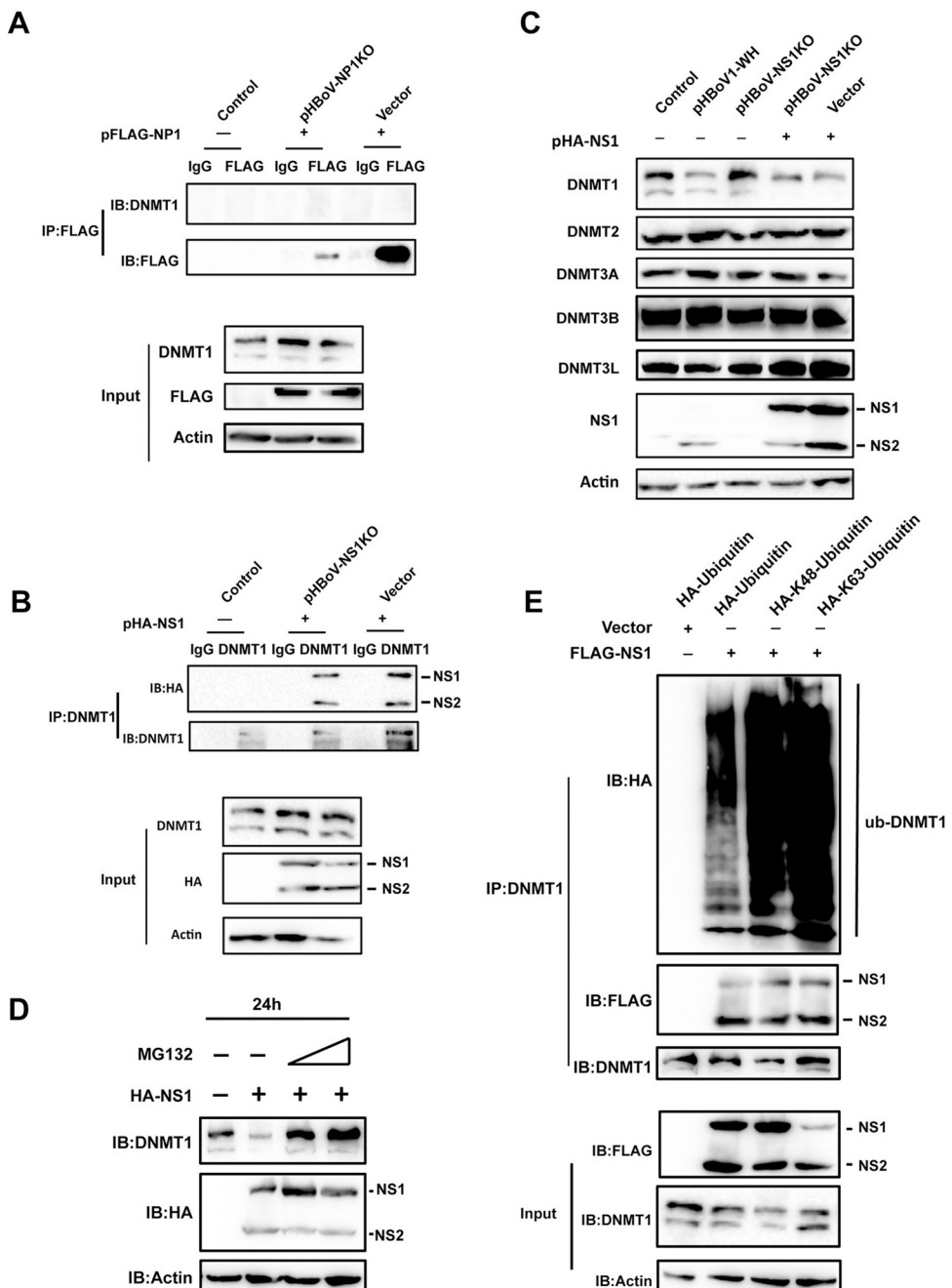

**Fig 5. NS1 promotes DNMT1 degradation through ubiquitin-proteasome pathway.** (A–B) DNMT1 interacted with NS1, not NP1. HEK293T cells were transfected with the indicated plasmids. Transfections with pHBoV-NP1KO and FLAG-NP1 (A), or pHBoV-NS1KO and HA-NS1 (B) were performed, respectively, to evaluate the interactions of NP1 or NS1 with DNMT1 in the presence HBoV transfection, while transfections with FLAG-NP1 (A) or HA-NS1 (B) only were performed to evaluate the interactions in the absence of HBoV transfection. The interactions were determined by co-IP experiments. (C) HBoV infection downregulated DNMT1 expression through NS1. HEK293T cells were transfected with the indicated plasmids. The expression of methyltransferases and NS1 were measured through western blotting. (D) NS1-mediated degradation of DNMT1 through the proteasome pathway. HEK293T cells were transfected with HA-NS1 and incubated with 1uM or 5uM MG132 for 24 h or 48 h. The expression of DNMT1 and NS1 were measured through western blotting. (E) NS1 promoted ubiquitination of DNMT1. HEK293T cells were transfected with FLAG-NS1 and HA-ubiquitin/HA-K48-ubiquitin/HA-K63-ubiquitin. DNMT1 was precipitated by the specific antibody followed by immunoblotting with the indicated antibodies.

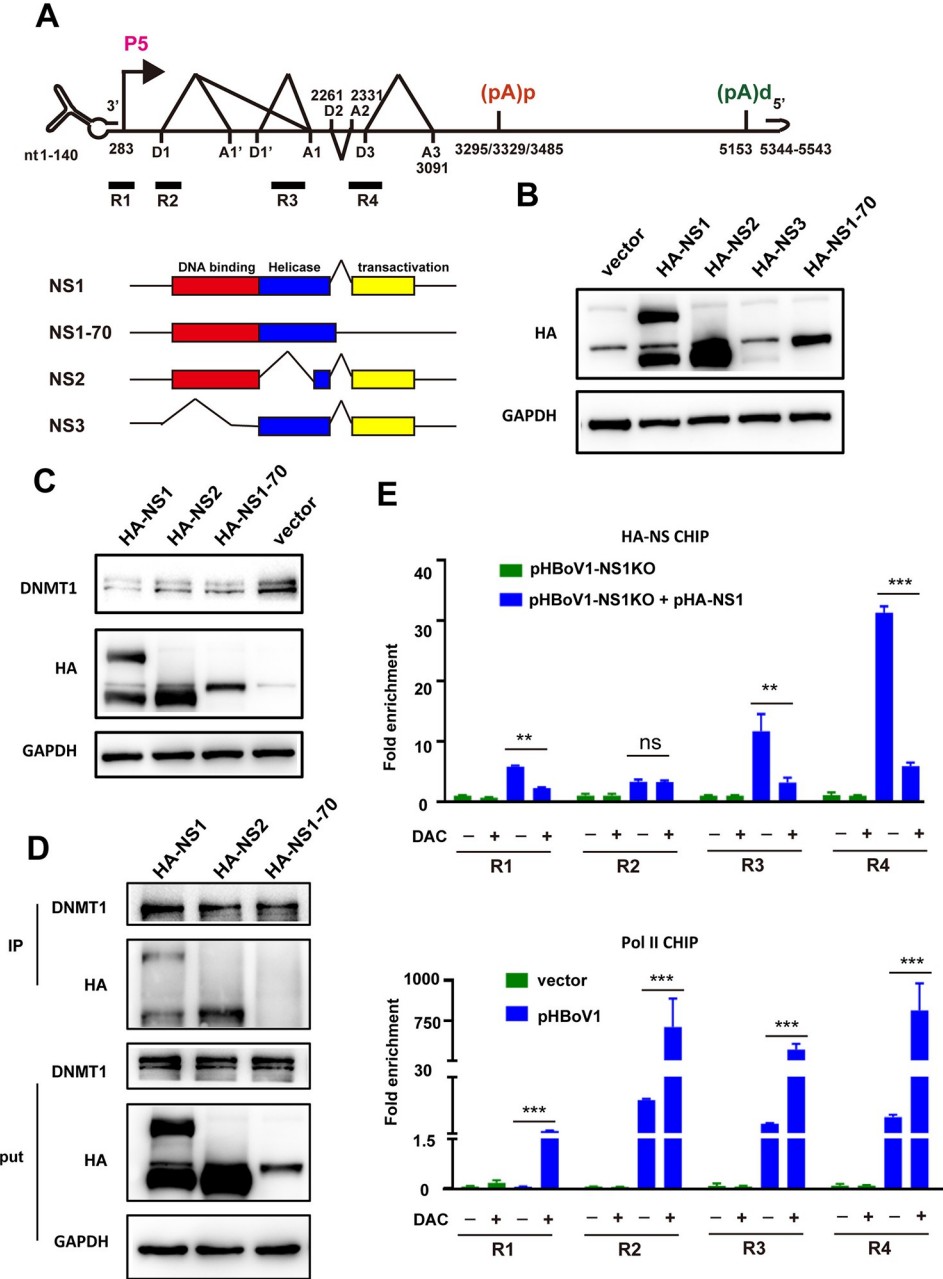

**Fig 6. Both NS1 and NS2 interacted with DNMT1 and promoted its degradation.** (A) A diagram of the NS isoforms is shown relative to the HBoV genome. (B and C) HEK293T cells were transfected with plasmids encoding HA-NS1, HA-NS2, HA-NS3, or HA-NS1-70. The expression levels of these isoforms were assessed by western blotting (B). Both HA-NS1, HA-NS2, and HA-NS1-70 were shown to promote DNMT1 degradation (C). (D) Co-immunoprecipitation (co-IP) was conducted using an anti-DNMT1 antibody at 48 hours post-transfection to evaluate the interactions with HA-NS1, HA-NS2, or HA-NS1-70. (E) HEK293T cells were transfected with the indicated plasmids and subsequently treated with DMSO or DAC. ChIP-qPCR was performed at 48 hours post-transfection using anti-HA or anti-Pol II antibodies, with four pairs of primers spanning the HBoV genome (A). **, P < 0.01; ***, P < 0.001.

knockdown cells have a synergistic effect on virus replication is unknown. As NS1 is essential for the recognition of replication origins and initiation of replication in the nucleus, cyto-plasm-localized NS1 may be the major cause of reduced viral replication. However, the

mechanism underlying the cytoplasmic retention of NS1 remains unclear. Inhibition of DNMT1 is also known to negatively affect host DNA synthesis and lead to decreased loading of the replication complex onto the replication origin by activating the ATR-dependent DNA damage response [53–55]. Thus, DNMT1 depletion may affect viral DNA replication in a similar manner. Interestingly, ATR- or ATM-dependent DNA damage responses are also induced by parvovirus infections and are important for viral DNA replication [56]. The NS1 protein, in the absence of viral genomic DNA, is sufficient to induce DNA damage responses in infections with parvovirus H-1 (H-1PV) and minute virus of mice (MVM) [56], which is attributed to the endonuclease activity of NS1 to create host DNA damage. NS1-mediated degradation of DNMT1 was slightly enhanced by both ATR inhibitor KU-60019 and ATM inhibitor VE-821 (S1G Fig), indicating a possible cross talk with virus induced DNA damage responses but not fully required for DNMT1 degradation. As ubiquitin-proteasome pathway is the most potential mechanism involved in the NS1-mediated DNMT1 degradation, further study of the E3 ligase involved enzymatic cascade is of interest in the future.

DNA methylation inactivates host gene expression by altering transcription factor binding [25], which is also observed in DNA viral infections [28,31–33]. Consistently, DNMT1 knockdown or inhibition of methylation by DAC resulted in increased viral RNA expression. Additionally, consistent with the function of DNA methylation in alternative RNA splicing [44] and polyadenylation [45], DNMT1 knockdown or methylation inhibition promoted viral RNA splicing and polyadenylation at the (pA)p site, which was associated with enhanced viral protein production, including non-structural and structural proteins (capsid protein). However, the NS1 expressed did not continue DNA synthesis when DNA methylation was inhibited because it was localized in the cytoplasm. This was further supported by the decreased NS1 recruitment on the viral DNA after DAC treatment (Fig 6E). Accordingly, when DNMT1 was degraded by NS1 after 48 h post-infection (Fig 1C and 1D), the viral genome was hypomethylated and released from methylation silencing for viral RNA expression and processing, especially those viral RNA transcripts producing capsid proteins. These findings also reveal a new function of NS1 that it not only initiates viral DNA replication, but also regulates HBoV RNA processing by promoting DNMT1 degradation.

Early NS1 and NP1 transcription is necessary for parvoviruses to initiate DNA replication. When viral DNA replication reaches an appropriate level, a certain amount of capsid production is required to initiate viral particle assembly. Our findings suggest that these processes are regulated by epigenetic mechanisms [26]. After the HBoV genomic DNA is uncoated in the nucleus, the hypermethylated HBoV genome maintains a low level of transcription to produce sufficient non-structural proteins, and the majority of the genomic DNA is used as a template for DNA synthesis. Under these conditions, DNMT1 was highly expressed in the nucleus and the methylation pattern was copied to the daughter DNA. Along with viral DNA replication, NS1 accumulated in the nucleus and induced DNMT1 degradation, which further resulted in hypomethylation of newly synthesized viral DNA (Fig 7). Thus, viral DNA is released from epigenetic silencing and begins structural protein transcription to produce a certain number of empty capsids for viral DNA genome packaging. This ultimately leads to the formation and release of new viral particles from the infected cells.

HBoVs cause lower respiratory tract diseases in young children. Although most infections cause only mild respiratory diseases, severe cases have also been reported [57]. Currently, there is no antiviral treatment for HBoV infections. Understanding viral replication mechanisms may provide new targets for the development of viral interference strategies. In this study, we demonstrate that DNA methylation and DNMT1 are important players in regulating HBoV replication and RNA processing. We also propose a new function for NS1, besides the initiation of DNA replication, it also regulates viral replication and RNA processing by

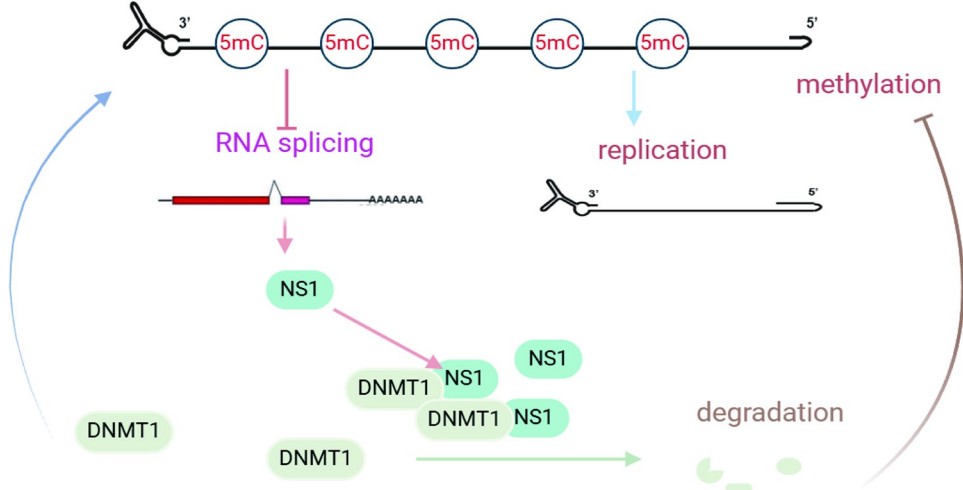

**Fig 7. NS1-mediated DNMT1 degradation regulates HBoV replication and RNA processing.**

promoting the degradation of DNMT1. Our study also indicated that targeting the DNA methylation system is a promising strategy for preventing HBoV infection.

## Limitations of the study

The results and conclusions were primarily based on the transfection of an HBoV infectious clone into HEK293T cells. In this transfection system, viral genome replication closely resembles that observed during natural infection, producing intermediate replication products such as double replicative form DNA, monomer replicative form DNA, and single-stranded DNA. Furthermore, the virus particles generated through transfection are infectious to polarized primary human airway epithelial cells, which are the natural host cells.

Although HEK293T cells do not support HBoV entry, they facilitate other aspects of the viral life cycle, including replication, RNA processing, and interactions involving DNA methylation and NS1-mediated DNMT1 degradation. These observations suggest that the mechanisms identified in HEK293T cells may represent a general regulatory process. However, further validation in an infection model is necessary to confirm these findings.

## Materials and methods

### Plasmid construction

The full-length HBoV infectious clone pHBoV1-WH, NP1 knockout plasmid pHBoV-NP1KO, and NS1 knockout plasmid pHBoV-NS1KO were constructed as described previously [40]. Both pXJ40-FLAG-NP1 and pXJ40-HA/FLAG-NS1 were generated by cloning NP1 and NS1 into the pXJ-40-FLAG/HA vector. pHA-ubiquitin, pHA-K48-ubiquitin, and pHA-K63-ubiquitin were gifted by Dr. Yanyi Wang (Wuhan Institute of Virology, Chinese Academy of Sciences, Wuhan, China). JM110 Chemically Competent Cell (Beijing Tsingke, TSC-C08) which were Dam and Dcm methylase deficient E. coli cells were used to amplify pHBoV1-WH plasmids.

### Cell lines and virus infection

HEK293T (ATCC, CRL-11268) cells were cultured in Dulbecco's modified Eagle's medium (Gibco) supplemented with 10% fetal bovine serum (Gibco) and 1× antibiotic-antimycotic solution (Gibco). All cells were cultured at 37˚C under humidified conditions with 5% $CO_2$.

Human bronchial epithelial Calu-3 cells were cultured in Dulbecco's modified Eagle's medium: F-12 medium (1:1) supplemented with 1% MEM nonessential amino acids solution (GIBCO), 1% GlutaMAX (GIBCO), and 10% fetal bovine serum (GIBCO) at 37°C under humidified conditions with 5% CO2, and the differentiated cells on semipermeable membrane inserts (0.6 cm$^2$; Millicell-PCF; Millipore) with transepithelial electrical resistance (TEER) values over 600 V/cm$^2$ were infected with HBoV at a multiplicity of infection (MOI) of 100 genome copy numbers/cell as described previously [40].

## Antibodies and reagents

Rabbit polyclonal antibodies against HBoV NP1 and NS1 were gifted by Prof. Hanzhong Wang. Mouse monoclonal antibodies against FLAG, HA, GAPDH, and beta-actin were purchased from Sigma (F1804), Proteintech (66006), Proteintech (60004), and Santa Cruz Biotechnology (sc47778), respectively. Rabbit polyclonal antibodies against β-tublin (Cell Signaling Technology, 2148), Pol II (abcam, EPR1509Y), histone H3 (GeneTex, GTX122148), DNMT1 (GeneTex, GTX116011), DNMT2 (Abcam, ab71015) DNMT3A (GeneTex, GTX129125), DNMT3B (GeneTex, GTX129127), and DNMT3L (GeneTex, GTX115985) were purchased. Other reagents such as Alexa Fluor 488 and Alexa Fluor 568-conjugated secondary antibodies (Invitrogen), 5-aza-2′-deoxycytidine (Sigma, A3656), T7 transcription kit (Ambion), RNase T1 (Ambion), RNase A (Ambion), MG132 (Beyotime), and Hoechst 33258 (Nalgene) were provided by the respective suppliers.

## Transfection and whole genome bisulfite sequencing

Transfection of 293T cells was performed using Lipofectamine 2000 (Invitrogen), according to the manufacturer's instructions. For whole-genome bisulfite sequencing, HEK293T cells were transfected with pHBoV1-WH for 48 h before low-molecular-weight (Hirt) DNA was isolated, and *Dpn*I digestion was performed [41,42]. Whole-genome bisulfite sequencing was performed by BGI Genomics, as described on the website https://www.yuque.com/yangyulan-ayaeq/oupzan/sdrdx4. Briefly, the isolated DNA was fragmented and sized. Methylated adaptors were then ligated to the DNA fragments by adding an A base to the 3' end. After bisulfite treatment and purification, the DNA was cyclized and amplified through rolling cycle amplification, loaded into patterned nanoarrays, and sequenced using combinatorial probe-anchor synthesis. Data were analyzed using a pipeline shown on the website https://www.yuque.com/yangyulan-ayaeq/oupzan/zpnmig.

## Western blot analysis and co-immunoprecipitation (co-IP)

For western blot analysis, total cellular proteins were dissolved in WB/IP buffer (50 mM Tris, pH = 7.5, 1 mM EGTA, 1 mM EDTA, 1% Triton X-100, 150 mM NaCl, 2 mM DTT, 100 μM PMSF, 1 μg/mL proteinase inhibitor) and quantified. Then, 30 μg total protein was denatured in SDS loading buffer at 95°C for 10 min and subjected to SDS-PAGE. The proteins were transferred to a nitrocellulose membrane, blocked, and incubated with the primary antibodies listed above, followed by incubation with secondary antibodies purchased from AntiGene Biotech GmbH. Luminescent signals were detected using a ChemiDocTM MP imaging system (Bio-Rad).

For co-IP, total protein was collected 48 h after transfection. Primary antibodies were mixed with supernatants of cell lysates (2 μg primary antibody per 1 mg protein sample) for 2 h at 4°C and then incubated with protein G agarose overnight at 4°C. Immunoprecipitated proteins were separated through SDS-PAGE on 12% gels and detected through western blotting.

## Southern blotting

To determine HBoV replication, low-molecular-weight (Hirt) DNA was isolated from cells with HBoV transfection and digested with *Dpn*I. Southern blot analysis of Hirt DNA was performed with DIG-labeled DNA probes (HBoV1 probe, nt 1–5543), as previously described [41,42]. DNA was purified using phenol:chloroform, separated on a 1% agarose gel, and transferred to a Hybond N+ membrane, followed by ultraviolet cross-linking. The membranes were hybridized with a HBoV1 probe (nt 1–5543) labeled with DIG using the DIG Luminescence Detection kit II (Roche, 11585614910) according to the manufacturer's protocol. Signals were detected using a ChemiDocTM MP imaging system (Bio-Rad).

## Northern blotting

Total RNA was extracted using the TRIzol reagent (Invitrogen). Northern blotting was performed as previously described [41,42]. In brief, 10 μg total RNA was dissolved and separated in 1.5% agarose gel containing 2.2 M formaldehyde. The RNA pellet was dissolved and run on a 1.5% agarose gel containing 2.2 M formaldehyde for 12 h at 28 V. The RNA was transferred to a Hybond-N$^+$ membrane by semi-dry transfer and then cross-linked using UV. Probe detection was performed using the DIG Luminescence Detection kit II (Roche, 11585614910) according to the manufacturer's protocol. Signals were detected using a ChemiDoc MP imaging system (Bio-Rad).

## RNA protection assay (RPA)

Total RNA was extracted using the TRIzol reagent. The RPA was performed as previously described [58]. Briefly, probes were produced from *in vitro* transcription of *Eco*RI-digested templates and labeled with [$^{32}$P]-GTP or [$^{32}$P]-CTP. Total RNA (10 μg) was protected with the indicated probes following RNase T1 and RNase A digestion. The reaction was terminated with proteinase K and samples were precipitated with absolute ethanol and glycogen at –80˚C. Then the samples were centrifuged, resuspended in 10 μL of 1× RNA loading buffer, and loaded onto a 6% urea denaturing gel before heating at 95˚C for 5 min. The RPA signals were detected and quantified using the Cyclone Storage Phosphor System (Packard Inc.) and analyzed using the OptiQuant Acquisition & Analysis software.

## Subcellular fractionation

HEK293T cells were transfected with 4 μg pHBoV1-WH for 48 h before subcellular fractionation. Cells were rinsed three times with cold phosphate-buffered saline (PBS) and then 400 μL homogenization buffer (10 mM HEPES, pH 7.4, 10 mM KCl, 1.5 mM MgCl2, 0.1 mM EGTA) was added to each 6-cm dish and collected with cell scrapers. Cell membranes were disrupted by passing through a 25 G needle three times and centrifuged 1000 g for 10 min at 4˚C. The supernatant was collected and centrifuged at 20,817 g for 30 min at 4˚C to get the purified cytoplasmic fraction. The nuclear fraction was washed three times with homogenization buffer. Extraction buffer (100 μL; 20 mM HEPES, pH 7.4, 0.4 M NaCI, 2 mM MgCl2, 1 mM EGTA, and 1 mM EDTA) was added to the pellet. The nuclear fraction was shaken in a cold room for 30 min before centrifugation and collection. The cytoplasmic and nuclear fractions were evaluated separately and mixed with 2 × loading buffer prior to western blotting.

## shRNA-mediated gene silencing

The shRNAs specific for DNMT1 (1#, 5'-CGACTACATCAAAGGCAGCAA-3', 2#, 5'-GCCCAATGAGACTGACATCAA-3') were cloned into pLKO.1-TRC clone vector (Addgene

plasmid 10878) and the lentiviruses were packaged by co-transfection with psPAX2 and pMD2.G into HEK293T cells. Stable knockdown cell lines were generated through standard viral infection on HEK293T cells and puromycin selection at 2 μg/mL.

### RNA extraction and quantitative real-time PCR (qRT-PCR)

Total RNA was extracted from the transfected cells using the TRIzol reagent according to the manufacturer's instructions. First-strand cDNA was synthesized using oligo (dT) 15 primers and M-MLV reverse transcriptase (Invitrogen). Quantitative real-time PCR was performed using the SYBR Green Real-time PCR Master Mix (Toyobo) on a Bio-Rad CFX96 (ABI). The copy NS1/NP1 mRNA copy numbers were normalized to GAPDH mRNA levels in each sample. For HBoV genome copy numbers, Hirt DNAs was isolated and *Dpn*I digestion was performed before quantification. Primers used were as follows: NP1 sense, 5'-AGAGGCTCGGG CTCATATCA-3', NP1 antisense, 5'-CACTTGGTCTGAGGTCTTCGAA-3'; NS1 sense, 5'-TGCAGACAACGCCTAGTTGTTT-3', NS1 antisense, 5'-CTGTCCCGCCCAAGATACA-3'; HBoV sense, 5'-AAGATACGCAACAAAGAAC-3', HBoV antisense, 5'-CAGGACTACAGT-CACCCT-3'; and GAPDH sense, 5'-GAAGGTGAAGGTCGGAGTC-3', GAPDH antisense, 5'-GAAGATGGTGATGGGATTTC-3'. All experiments were performed in triplicates.

### Immunofluorescence (IF)

To determine the subcellular locations of NP1 and NS1, indirect IF was performed as follows: Cells were plated in a 12-well plate one day before transfection. Media were changed six hours post transfection (h.p.t); then, DMSO or DAC was added to each well. After 42 h, the cells were fixed in 4% paraformaldehyde overnight, permeabilized in 0.2% Triton X-100 for 10 min, washed three times with PBS, and blocked with 3% bovine serum albumin for 1 h at room temperature. The cells were incubated with primary antibodies overnight at 4˚C at the dilution suggested by the manufacturer, and stained with secondary antibodies (Alexa Fluor 488 and Alexa Fluor 568) for 1 h after three washes with PBS. Nuclei were visualized using Hoechst 33258 at a dilution of 1:1,000. Finally, the cells were observed under a PE VoX confocal microscope.

### Chromatin immunoprecipitation (ChIP)

HEK293T cells were transfected with indicated plasmids, and treated with DAC as described above, and then the CHIP experiments were performed with ChIP-IT High-Sensitivity (HS) Kit (Active Motif, 53040) as described previously [59]. The sheared chromatin was analyzed in agarose gel to confirm the DNA fragments at the appropriate size (200–600 bp) before CHIP assay. The immunoprecipitated was then performed with anti-HA or anti-Pol II antibodies overnight at 4˚C. The recovered DNA from the protein G agarose was analyzed by qPCR with four pair of primers.

The sequences of qPCR primers: Region 1 (5'-GATCTAATCGCCGGCAGACA-3', 5'-AGCGTGGAGCTTTTTCCAGA-3'); Region 2 (5'-TCTGGAAAAAGCTCCACGCT-3', 5'-TTGCAGAGAACAGCCCCAAA-3'); Region 3 (5'-TTGGGTGGAACCTGCAAAGT-3', 5'-TAATGGAGCCGCGTGAACAT-3'); Region 4 (5'-GAGCCTGAGACATCGCAAGT-3',5'-ATATGAGCCCGAGCCTCTCT-3').

### Statistical analysis

Statistical analyses were performed using one-way analysis of variance (ANOVA) for multiple comparisons followed by Turkey's post-test analysis with GraphPad Prism Software. Statistical

significance was set at $p < 0.05$. Data are presented as the mean ± standard deviation (SD). All experiments were performed in triplicate.

## Supporting information

**S1 Fig.** (A-B) The Dpn I digested Hirt DNAs (A) and the mRNAs expressing NS1 or NP1 (B) were quantified and analyzed through qRT-PCR in Calu-3 cells after DAC treatment. GAPDH was used as a control. *, $p < 0.05$, **, $p < 0.01$. (C) HEK293T cells were transfected with pHBoV1-WH and then treated with DMSO or 20uM DAC at 48h post-transfection, and the cell cycle of the treated cells were then analyzed by propidium iodide (PI) flow cytometry assay. (D) HEK293T cells were transfected with pHBoV-NS1KO and HA-NS1 and treated with DAC. To visualize the NS1, HA-NS1 was detected by IF using specific antibody against HA (green). The nuclei were stained with Hoechst 33258 (blue). The nuclear border is labeled by a white dashed line. (E) HEK293T cells with Stably knockdown of DNMT were selected as described in Materials and Methods. The cells were then transfected with pHBoV1-WH, and the cell cycle was analyzed by propidium iodide flow cytometry assay. (F) Subcellular fraction-ation of DNMT1 knockdown cells transfected with pHBoV-NS1KO and HA-NS1 was per-formed to validate the localization of HA-NS1 by western blot. The subcellular fractions of β-tublin and histone H3 indicated the efficient subcellular fractionation. (G) HEK293T cells were transfected with HA-NS1 and incubated with 20uM or 30uM KU-60019 or VE-821 (APE×BIO) for 24 h. The expression of DNMT1 and NS1 were measured by western blotting. (TIF)

**S1 Table. HBoV methylation sites after DpnI digestion.**
(DOCX)

## Acknowledgments

We thank the Core Facility and Technical Support in the Wuhan Institute of Virology (WIV), Chinese Academy of Sciences (CAS), especially Ding Gao, for the assistance with confocal and isotope experiments.

## Author Contributions

**Conceptualization:** Haibin Liu, Xiulian Sun, Wuxiang Guan.

**Data curation:** Shuangkang Qin, Honghe Chen.

**Formal analysis:** Shuangkang Qin, Honghe Chen, Zhen Chen, Li Zuo, Xueyan Zhang, Haojie Hao, Fang Huang.

**Funding acquisition:** Honghe Chen, Wuxiang Guan.

**Investigation:** Shuangkang Qin, Honghe Chen, Chuchu Tian, Zhen Chen.

**Supervision:** Wuxiang Guan.

**Writing – original draft:** Honghe Chen, Haibin Liu, Wuxiang Guan.

**Writing – review & editing:** Shuangkang Qin, Honghe Chen, Chuchu Tian, Zhen Chen, Li Zuo, Xueyan Zhang, Haojie Hao, Fang Huang, Haibin Liu, Xiulian Sun, Wuxiang Guan.

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
