## [Decision Letter · Decision Letter 0]

20 Oct 2024

Dear Dr. Guan,

We are pleased to inform you that your manuscript 'NS1-mediated DNMT1 degradation regulates human bocavirus 1 replication and RNA processing' has been provisionally accepted for publication in PLOS Pathogens.

Best regards,

Cary A. Moody

Academic Editor

PLOS Pathogens

Alison McBride

Section Editor

PLOS Pathogens

Michael Malim

Editor-in-Chief

PLOS Pathogens

orcid.org/0000-0002-7699-2064

Reviewer Comments (if any, and for reference):

Reviewer's Responses to Questions

**Part I - Summary**

Reviewer #1: In this study, Qin et al. demonstrate that DNA methylation of the HBoV genome by DNMT1 regulates virus replication and viral gene expression. The authors have appropriately addressed all of the major points raised by reviewers in the first submission, going above and beyond by performing additional experiments to validate their model. These results have addressed the major concerns that were previously raised and improved the excitement of their findings. The overall findings of this study are applicable to understanding the epigenetic regulation of viral life cycles in general and parvovirus life cycle in particular.

Reviewer #2: Dear Dr. Moody,

As stated in my initial review I appreciate the opportunity to participate in the review of the Qin et al. manuscript (NS1-mediated DNMT1 degradation regulates human bocavirus 1 replication and RNA processing). Based on the results of the additional experiments and new insights into the mechanism associated with NS1-mediated DNMT1 degradation in human bocavirus 1 replication and RNA processing, I recommend and approve the manuscript for publication in Plospathogen.

Strengths/Novelty/Significance:

•Qin et al. utilized the HBoV model to characterize the role of DNA methylation in parvovirus DNA replication and RNA processing.

•The study effectively characterized and showed that DNA methylation inhibition with 5-aza-2′-deoxycytidine (DAC) in HEK293T cells and Calu-3 cells coupled DNMT1 shRNA knockdown sufficiently revealed alteration in parvoviral DNA replication and RNA processing.

•Similarly, DNA methylation inhibition with 5-aza-2′-deoxycytidine (DAC) in HEK293T cells and Calu-3 cells coupled DNMT1 shRNA knockdown indicated altered expression of NS1 and NP1 protein in addition to predominant NS1 cytoplasmic localization.

•Immunoprecipitation of DNMT1 with NS1 added credence to the potential role of DNA methyltransferase in parvoviral gene expression and host-cell interaction.

•The result provided evidence that NS1 expressed from the infectious clone pHBoV1-WH and epitope-tag construct (HA-NS1 and FLAG-NS1) modulated the DNMT1 depletion with restoration after MG132 treatment in a proteasome-dependent manner.

•The study explores novel questions that have not been comprehensively explored in parvoviral biology and host-cell interaction, with potential implications for the advancement of parvoviral gene therapy.

•This study will expand the frontiers of parvovirus and DNA viruses host cell interaction.

Sincerely,

Olufemi Fasina DVM, Ph.D., Diplomate. ACVP

Assistant Professor

Department of Veterinary Pathology

2720 College of Veterinary Medicine

Iowa State University

515-294-5227

ofasina1@iastate.edu

**Part II – Major Issues: Key Experiments Required for Acceptance**

Reviewer #1: None noted

Reviewer #2: The weakness raised in the initial review have been addressed.

**Part III – Minor Issues: Editorial and Data Presentation Modifications**

Reviewer #1: None noted

Reviewer #2: The editorial and data presentation modifications raised in the initial review have been addressed.

PLOS authors have the option to publish the peer review history of their article (what does this mean?). If published, this will include your full peer review and any attached files.

Reviewer #1: No

Reviewer #2: **Yes: **Olufemi Fasina DVM, Ph.D., Diplomate. ACVP

---

## [Editor Report · Acceptance letter]

7 Nov 2024

Dear Dr. Guan,

We are delighted to inform you that your manuscript, " NS1-mediated DNMT1 degradation regulates human bocavirus 1 replication and RNA processing ," has been formally accepted for publication in PLOS Pathogens.

Best regards,

Michael Malim

Editor-in-Chief

PLOS Pathogens

orcid.org/0000-0002-7699-2064